# Minimising the Toxicities of First Line Hodgkin Lymphoma Treatment in the Modern Era

**DOI:** 10.3390/cancers14215390

**Published:** 2022-11-01

**Authors:** Annabel M. Follows, Anna Santarsieri

**Affiliations:** 1University of Cambridge School of Clinical Medicine, Cambridge Biomedical Campus, Cambridge CB2 0SP, UK; 2Department of Haematology, Cambridge University Hospitals NHS Foundation Trust, Cambridge CB2 0QQ, UK

**Keywords:** Hodgkin lymphoma, chemotherapy, radiotherapy, toxicity

## Abstract

**Simple Summary:**

Due to advances in chemotherapy and radiotherapy over recent decades, Hodgkin lymphoma is now a curable disease in the vast majority of younger patients. While treatment has become very effective, patients are burdened with significant short- and long-term side effects such as infections, infertility, organ failure and secondary malignancies. In this article we review the strategies that have been used to reduce the toxicity of treatment while maintaining high cure rates, and how the current treatment of Hodgkin lymphoma has evolved based on the success of these strategies.

**Abstract:**

Striking advances in the treatment of Hodgkin lymphoma over the last 30 years have culminated in high rates of disease-free survival in younger patients with early and advanced stage disease. In this review we focus on strategies that have evolved over recent years to reduce short and long-term toxicities of treatment. These strategies include the selection of first-line chemotherapy, the stratification of patients based on initial response and subsequent adaptation of treatment, the addition of novel agents (e.g., brentuximab vedotin), the removal of specific drugs (e.g., bleomycin), the use of drug substitution, and the removal of consolidation radiotherapy based on interim and end of treatment PET assessment. While these strategies have successfully reduced toxicity of Hodgkin lymphoma therapy, the cornerstone of treatment continues to be combination chemotherapy and radiotherapy with significant short- and long-term side effects. To further reduce toxicity while maintaining or improving efficacy, we shall need to incorporate novel agents into our first-line treatment algorithms, and several such potentially practice-changing trials are underway.

## 1. Introduction

The management of Hodgkin Lymphoma (HL) is an outstanding example of how modern oncological treatment has improved over the last 30 years to the point where we now hope to cure almost all younger patients who present with this form of cancer. As treatment has evolved from radiotherapy alone to combined modality chemotherapy/radiotherapy and increasingly chemotherapy without radiotherapy consolidation, 5-year overall survival (OS) rates in excess of 95% are now expected when treating younger patients with either early- and advanced-stage disease [1]. Although the long-term survival rates in patients aged over 60 remain inferior [2], recent data suggest this is also now improving, so for young and old patients alike, there is an imperative to develop treatments that not only maximise efficacy, but also minimise both the short and long-term toxicities of treatment. The dominant short-term toxicities reflect different chemotherapy drugs which cause acute toxicity and carry the small but significant risk of treatment-related mortality, must typically as a result of neutropenic sepsis [3]. In the longer term, chemotherapy and radiotherapy can both contribute to organ dysfunction, infertility and increase the risk of secondary malignancies [4,5,6]. Indeed, a recent multicentre study showed that most long-term survivors of HL eventually die from causes likely related to their previous HL treatment [7]. Clearly a balance needs to be struck; a first line treatment that compromises efficacy to reduce toxicity will leave a patient exposed to the additional risks associated with second-line therapy, including intensive chemotherapy delivered as part of an autologous stem cell transplant [4]. The ultimate goal for oncologists and patients alike is therefore to balance maximal efficacy for each individual patient while minimising the risk of short and long-term toxicities.

This review will consider several strategies employed to reduce the toxicity burden when treating early and advanced-stage HL. Broadly speaking, these strategies aim to optimise treatment for all patients primarily by intensifying the treatment of poorer risk patients and de-escalating treatment intensity for better risk patients. These decisions start with the selection of initial chemotherapy and continue with de-escalation or intensification strategies based on interim response assessments. Strategies to improve the durability of first line remissions include intensification of the initial chemotherapy through the use of more complex and intensive regimens such as escalated bleomycin, etoposide, doxorubicin, cyclophosphamide, vincristine, prednisolone, procarbazine (eBEACOPP) and for specific patients through the addition of newer targeted drugs such as brentuximab vedotin (BV). There is potential to reduce toxicity through drug substitution, as exemplified in paediatric practice where the gonado-toxic drug procarbazine is commonly replaced with the less toxic dacarbazine [8]. Weight of evidence now supports the use of interim positron emission tomography (iPET) scanning to identify more accurately patients who are doing well, or less well, with initial cycles of treatment. This may permit the removal of specific drugs, such as bleomycin, from subsequent treatment chemotherapy cycles for good risk patients and may allow the total number of chemotherapy cycles delivered to be reduced. Furthermore, iPET assessment may help guide difficult decisions to remove radiotherapy consolidation post chemotherapy. For patients doing less well at iPET assessment, there is increasing evidence for those treated with doxorubicin, bleomycin, vinblastine, and dacarbazine (ABVD), that treatment intensification with more intensive chemotherapy and radiotherapy can reduce the risk of relapse after first line treatment and reduce the subsequent need for salvage therapy. Alongside all of these strategies to reduce the toxicities of physical treatment, the impact of psychological toxicity cannot be underestimated, and this risk is accentuated by prolonged periods of treatment and the need for relapse treatment, particularly in young cancer patients [9,10].

## 2. Early-Stage Favourable Hodgkin Lymphoma

Early-stage HL patients can present with either favourable or unfavourable disease. International study groups have slightly different definitions of these two criteria (Table 1). The German Hodgkin Study Group (GHSG) criteria for favourable disease are more stringent in terms of lymph node (LN) groups (≤2: as illustrated in Figure 1) but permit older patients to be classified as ‘favourable’. The European Organisation for Research and Treatment of Cancer (EORTC) and National Cancer Institute of Canada/Eastern Cooperative Oncology Group (NCIC/ECOG) criteria permit favourable patients to have up to three involved LN groups but patients must be under the age of 50 to be considered good risk. There are also minor differences in erythrocyte sedimentation rate (ESR) cut-off points. For stage II patients who present with B symptoms, the German group does not consider these as early-stage if they have either a bulky mediastinal mass or extranodal disease. This is important, as these patients should be managed with advanced-stage protocols.

Favourable early-stage HL has an excellent prognosis, when treated with ABVD chemotherapy followed by radiotherapy consolidation. The landmark prospective randomised GHSG HD10 trial demonstrated clearly that ABVD could be reduced from 4 cycles to 2 cycles and involved field radiotherapy (IFRT) reduced from 30 Gy to 20 Gy without loss of efficacy [11]. This defined 2 cycles of ABVD and 20Gy IFRT as the international standard of care for these best risk patients. There are, however, significant toxicities associated ABVD. The Italian GISL study identified the most common grade 3–4 acute adverse effects as neutropenia (34%), alopecia (31%), nausea/vomiting (13%), anaemia (5%), thrombocytopenia (3%), infections (2%), constipation (2%), and mucositis (1%) [12]. Longer-term toxicities of ABVD were tracked over time in a NCIC/ECOG trial, and the most commonly noted were pulmonary toxicity (8%), cardiac dysfunction (3%), and secondary malignancies (4%) [13]. Within the ABVD regimen, bleomycin in particular has been linked to significant morbidity through pulmonary toxicity. Indeed, in one study of advanced-stage HL, 24% of older patients developed bleomycin lung toxicity (BLT), with a BLT-related mortality rate of 18% [14]. There has therefore been considerable interest in whether specific drugs could be removed from ABVD to improve safety without compromising efficacy. The GHSG HD13 trial prospectively addressed this question by randomly assigning good risk early-stage patients to treatment with or without bleomycin and/or dacarbazine [15]. All patients then received 30Gy IFRT. Unfortunately, high rates of progression or early relapse meant both groups where dacarbazine was omitted (ABV and AV) had to be prematurely stopped, concluding that dacarbazine cannot be omitted without significant loss of efficacy. In the group where bleomycin alone was omitted (AVD), tumour control was marginally inferior to the control group (ABVD) with a 3.9% deficit in freedom from treatment failure, although OS did not differ between the two groups. However, as the predefined inferiority margins were not met, the trial concluded that bleomycin could also not be safely omitted, leaving 2 cycles of ABVD as the standard chemotherapy regimen of choice for these patients. Pragmatically, with certain patients where the risk of bleomycin lung toxicity is elevated [16] (elderly, long-term smokers, renal impairment, cumulative bleomycin dose, etc.), this potentially fatal risk may exceed the progression free survival (PFS) deficit from bleomycin omission, so treating with AVD will remain appropriate for specific individuals [17]. Indeed, it is now common practice in the UK to remove bleomycin from ABVD, either initially or after 2 cycles of ABVD, when treating patients over 60 years [17].

With many early-stage patients presenting at a young age there has been considerable interest in developing radiation-free strategies to minimise longer-term treatment risks. This is particularly important as young patients commonly present with mediastinal disease and radiation fields will typically include cardiac structures such as the coronary ostia and breast tissue [18]. The first large prospective trial to explore radiotherapy-free treatment in good risk patients was the Canadian HD6 study which randomised patients to either extended-field radiotherapy alone or ABVD delivered as 4 or 6 cycles without radiotherapy consolidation [19]. This trial was not PET-stratified, and after 2 cycles of ABVD, patients who had achieved a complete remission by computed tomography (CT) scan criteria were given 4 cycles of ABVD in total, while PR patients received 6 cycles of ABVD in total. When an interim report was presented after a median follow-up of 4.2 years the ABVD-only group had a significantly worse rate of freedom from progression (87% vs. 93%; *p* = 0.006) [20]. Clearly radiotherapy reduces the risk of early relapse and this has now been confirmed by multiple later trials in early-stage HL [21,22,23]. However, with 11.3 years of median follow-up, there were no relapses beyond 5 years in the ABVD-treated patients and the long-term OS in the radiotherapy arm was statistically inferior to the ABVD-only arm (87% vs. 94%; *p* = 0.04). While the reasons behind this OS inferiority have been hotly debated, a key take-home message from this trial was that ABVD alone could cure the vast majority of young, good risk early-stage HL patients. However, the majority of these chemotherapy patients were treated with 6 cycles of ABVD and there are significant toxicities associated with additional cycles of ABVD [24].

The Canadian HD6 trial was not PET-stratified, but three large prospective randomised trials have now explored whether iPET rather than CT assessments can provide sufficient response assessment to predict which good risk patients could be treated with fewer cycles of ABVD and without radiotherapy [21,22,23]. In the UK National Cancer Research Institute (NCRI) RAPID trial [21] early-stage favourable patients received 3 cycles of ABVD, before a PET scan. Patients who achieved a complete metabolic response (CMR), were randomly assigned to either no further treatment (NFT) or to IFRT. All patients who failed to achieve a CMR were given an additional cycle of ABVD and IFRT. For CMR patients, the 3-year PFS was inferior for the NFT rather than IFRT (90.8% vs. 97.1%, *p* = 0.02 in the per-protocol analysis). In the HD16 trial, all patients received 2 cycles of ABVD prior to a PET scan [23]. In the experimental arm, CMR patients did not receive radiotherapy, whereas in the standard arm all patients received 20Gy IFRT independent of the iPET2 result. Again, with 45 months median follow-up, the PFS was significantly inferior for the radiation-free arm compared to the control arm (86.1% vs. 93.4%; hazard ratio (HR) = 1.78; [95% confidence interval (CI), 1.02–3.12]). The H10 trial design was more complecycles of with all patients starting with 2 cycles of ABVD followed by an iPET2 scan [22]. With the favourable standard arm treatment was 3 cycles of ABVD and involved node radiotherapy (INRT), i.e., iPET2 performed, but not used to plan treatment. The experimental arm was iPET2 stratified, with CMR patients receiving 2 cycles of further ABVD while non-CMR patients received 2 cycles of eBEACOPP and INRT. Again, radiotherapy consolidation improved immediate disease control in the iPET2 CMR group, with 5-year PFS rates of 99.0% for 3 cycles of ABVD and INRT vs. 87.1% for 4 cycles of ABVD (HR = 15.8; [95% CI, 3.8–66.1]). For patients who did not achieve CMR, there was a progression free survival benefit for patients who received 2 cycles of eBEACOPP rather than 1 cycle of ABVD post iPET and prior to radiotherapy (5-year PFS improved from 77.4% for standard ABVD and INRT to 90.6% for intensification to eBEACOPP and INRT (HR = 0.42; [95% CI, 0.23–0.74]; *p* = 0.002).

There are consistent noteworthy features across all three iPET-directed trials. Firstly, removing radiotherapy consolidation from iPET CMR patients increases the relapse risk, i.e., we cannot use iPET to reliably identify patients who do not need radiotherapy consolidation. Secondly, whether or not radiotherapy is used, the 5-year OS approached 100% for younger patients who achieved a CMR after 2 or 3 cycles of ABVD. Thirdly, patients who failed to achieve a CMR after 2 or 3 cycles of ABVD have an inferior progression free and OS despite the consistent use of radiotherapy for these patients in all three trials. For these poorer risk patients, the H10 trial was the first prospective randomised trial to show that intensification of ABVD to eBEACOPP in non-CMR patients reduces the risk of relapse and indeed likely improves OS (5-year OS 89.3% vs. 96.0%, HR = 0.45; [95% CI, 0.19–1.07]; *p* = 0.062). Finally, the shrinking of radiotherapy fields from extended field to involved field to involved node has significantly reduced the extent of the exposed radiotherapy field and while these changes have not been validated in a prospective randomised trial, there is no evidence to suggest a loss of disease control with these more targeted techniques [17].

So where does this leave the clinician and patient deciding on a treatment plan for a patient with early-stage good risk HL? If a patient is fit enough for anthracycline-based therapy, then 2 cycles of ABVD followed by an iPET will be the standard of care for most patients. Choosing to omit the bleomycin where there are specific bleomycin risk factors, such as age, smoking, renal impairment, will be a personalised decision. If patients achieve a CMR, then going directly to radiotherapy consolidation (as per HD16) or after one further ABVD (as per H10) may depend on the tolerance of ABVD and whether the patient met the stricter German LN criteria for early-stage favourable disease at initial presentation. However, some younger patients may specifically chose not to be treated with radiotherapy because of concern for longer term risks. It is important that each case is discussed carefully with a radiation oncologist so the extent of the radiotherapy field and these risks can be best understood. If the decision is made to omit the radiotherapy, then the prospective trials show us that around 1 in 10 patients will relapse specifically because of this decision. Many of these patients will then require more intensive chemotherapy, including an autologous stem cell transplant which comes with significant morbidity and an additional set of longer-term risks including infertility and secondary malignancies [25]. A further difficulty is that if a patient declines radiotherapy, then the optimal number of ABVD cycles to deliver remains unclear, although UK guidelines suggest at least 3 cycles [17]. For many patients who fail to achieve a CMR after 2 cycles of ABVD the best strategy to reduce the longer-term risks of relapse would be to intensify therapy to 2 cycles of eBEACOPP prior to radiotherapy. For these patients, omitting radiotherapy has not been tested in a randomised trial, but noting the results from the better risk CMR patients, it is likely that omitting radiotherapy from this in this higher risk group would have a disproportionate impact on failure free survival.

## 3. Early-Stage Unfavourable Hodgkin Lymphoma

Early-stage HL with unfavourable prognostic factors is a complex area for both clinicians and patients. Standard protocols have historically included radiotherapy for all patients, which by the nature of the initial presentation, mean radiotherapy fields are often large, raising concerns for longer-term risks. Many clinicians have tended to treat these patients with advanced-stage protocols, as shown by the UK Risk-Adapted Therapy in HL (RATHL) trial, where 41.6% of patients in this trial, which was designed for advanced-stage HL, were actually stage II patients with poorer risk features [26]. The GHSG excludes stage II bulky patients who have B symptoms or extra-nodal disease from early-stage trials, and 14% of the patients treated in the advanced-stage HD18 were stage II patients [27]. For patients, the term “early-stage unfavourable” can be challenging, thus “early-stage intermediate” is increasingly used in the UK [17].

Historically, the GHSG HD14 trial had defined a standard of care for this group of patients as 2 cycles of eBEACOPP and 2 cycles of ABVD with 30Gy IFRT with 5-year PFS of 95.4% [28]. However, the additional toxicity of this protocol compared with 4 cycles of ABVD and 30 Gy IFRT meant this standard was not adopted widely outside the GHSG. Interestingly however, recent longer-term findings of the HD14 trial dispute the earlier conclusions of additional toxicity of the “2 + 2” regimen, suggesting the need for reconsideration of this trial in the context of its recent findings [29].

Alongside this new update from the HD14 follow-up, a key question for unfavourable HL treatment, as with favourable HL, has been whether iPET assessment can help stratify patients and allow treatment intensity to be reduced, particularly in good responders, through removal of radiotherapy. The GHSG HD17 trial and EORTC H10 have approached this question with different strategies [30]. With both trials, control patients were allocated the appropriate standard of care including radiotherapy as consolidation irrespective of results from an interim/post-chemotherapy PET scan, while the experimental arms were managed beyond the PET stratification point with CMR patients receiving no radiotherapy. With HD17, the PET was performed after 4 cycles of chemotherapy (2 cycles of eBEACOPP and 2 cycles of ABVD), i.e., after all chemotherapy was completed. The results showed that patients in the experimental arm, where radiotherapy was given or not based on the post-chemotherapy PET scan, had a non-inferior 5-year PFS compared with patients treated in the control arm who all received 30 Gy INRT (95.1% vs. 97.3%; HR = 0.523; [95% CI, 0.226–1.211]). Approximately 90% of patients are able to achieve CMR at this point, meaning the clear majority of patients with early-stage unfavourable HL can be treated without radiotherapy using this protocol and achieve excellent long-term disease-free survival without adding additional longer-term risks of radiotherapy.

The EORTC H10 trial took a different approach and started all patients with 2 cycles of ABVD prior to the iPET [22]. The control arm then received 2 cycles of further ABVD and 30 Gy INRT. In the experimental arm, the CMR patients received 4 further cycles of ABVD, i.e., 6 cycles in total, without radiotherapy, while patients who failed to achieve a CMR were intensified with 2 cycles of BEACOPP and 30 Gy INRT. Non-inferiority could not be demonstrated, although treatment results between the experimental and standard arms for patients with a CMR at iPET were very similar, with 5-year PFS not differing significantly between the standard arm (total 4 cycles of ABVD and INRT) and the experimental arm (6 cycles of ABVD with no radiotherapy) (92.1% vs. 89.6%, respectively; HR = 1.45; [95% CI, 0.8–2.5]). For the H10 trial as a whole, the non-CMR patients who continued with ABVD (4 cycles in total followed by INRT) had a relatively poor PFS compared with patients escalated to eBEACOPP cycles of 2 prior to 30 Gy INRT (77.4% vs. 90.6%; HR = 0.42; [95% CI, 0.23–0.74]; *p* = 0.002).

GHSG HD17 and the EORTC H10 trials have therefore given patients and clinicians a choice of 2 new standards of care, with both protocols using iPET to stratify whether or not consolidation radiotherapy is required [22,30]. Starting with an HD17 approach will mean all patients receive 2 cycles of more intensive eBEACOPP chemotherapy, whereas starting with ABVD will mean only a minority of patients will need to be given this more intensive chemotherapy. However, starting with ABVD will mean even a good risk iPET CMR patient will receive 24 weeks of chemotherapy as opposed to 14 weeks with HD17. Furthermore, starting with ABVD, 21% of early-stage unfavourable patients did not achieve CMR in the H10 trial and therefore 1 in 5 patients starting with ABVD will require chemotherapy intensification and radiotherapy treatment. Prior to starting treatment, clinicians and patients need open conversations as to where patient preferences lie. For some patients, concern for potential toxicities with eBEACOPP may be more troubling than 24 weeks of less intensive chemotherapy with ABVD. This may direct them to an H10 approach. For others, shorter course more intensive chemotherapy treatment with a relatively low risk of needing radiotherapy my push them towards HD17. It is noteworthy that a separate study surveyed patient preferences for treatment characteristics associated with frontline HL therapies, with patients reporting that they were likely to accept treatments with higher risks of adverse events if the risk of relapse was lower [31]. This emphasises the importance of patient preference and furthermore reminds clinicians of the potential psychological toxicities for patients from both the initial treatment given and the fear and reality of relapse if it happens. Clinicians should be careful not to assume they know the values and preferences most important to an individual patient, and for this reason, the UK Hodgkin Lymphoma Guidelines retain both H10 and HD17 strategies as standards of care for early-stage unfavourable patients [17]. One area that has prompted on-going debate in the UK is the timing of the iPET scan for early-stage unfavourable patients treated with an HD17 approach. If a clear priority is to avoid radiotherapy, some clinicians are opting for iPET after 2 cycles of eBEACOPP, i.e., following a similar initial strategy to advanced-stage disease [32]. If patients achieve a CMR on this scan, then following the standard protocol with 2 cycles of ABVD and no radiotherapy is reasonable. If patients do not achieve iPET CMR, then completing a total of 6 cycles of eBEACOPP, as per the HD18 protocol may be a reasonable option for some patients in a further effort to try and avoid radiotherapy. This strategy is not, however, validated by prospective trial data.

Finally, it is important to briefly note recent studies assessing the role of brentuximab vedotin (BV), a novel and targeted drug, in the context of early-stage unfavourable HL. The phase II BREACH trial assessed the efficacy and safety of BV in combination with AVD (BV-AVD), observing an increased rate of negative PET scans after 2 cycles of BV-AVD compared with 2 cycles of ABVD [33]. While still in an early phase, studies such as this show the potential to be practice-changing trial, emphasising the important role of novel agents in developing highly effective but minimally toxic treatments.

## 4. Advanced-Stage Hodgkin Lymphoma

Advanced-stage HL includes patients with stage III or IV disease, but patients with stage II bulky mediastinal disease who present with B symptoms or extranodal disease are usually managed with advanced-stage protocols [17]. As with early-stage disease, the clear majority of patients will be cured after first line treatment, so optimising therapy to reduce short- and longer-term toxicities is a priority. Historically, advanced-stage patients, and particularly those with a higher international prognostic score (IPS) have had a worse prognosis, and therefore reducing treatment intensity, particularly in these poorer risk patients needs to be done with caution. The two most widely used regimens to treated advanced-stage disease are ABVD and eBEACOPP. While there is general international consensus that eBEACOPP is more effective at treating HL and inducing remissions with a reduced risk of relapse, ABVD is generally less toxic in both the short and long-term [12,34,35]. Indeed, the GHSG do not recommend eBEACOPP for patients aged over 60 unless there are very persuasive reasons for its use.

A number of clinical trials have directly compared ABVD and eBEACOPP in an advanced-stage HL setting [12,35,36,37]. Although trial designs have varied, none of the head-to-head comparative trials have included iPET assessments to stratify patients and modify treatment based on an interim response. The two trials with largest PFS differences between the 2 regimens were the Lymphoma study association (LYSA) H34 trial in better risk IPS patients and the HD2000 trial in standard risk [12,37]. With LYSA H34 there was a superior 5-year PFS with 6 cycles of eBEACOPP compared with 6 cycles of ABVD (93% vs. 75%, *p* = 0.008) and with HD2000 the 5-year PFS with 4 cycles of eBEACOPP and 2 cycles of standard BEACOPP compared with 6 cycles of ABVD was 81% vs. 68% (*p* = 0.038). With the exception of LYSA H34, it has been hard to show a statistical improvement in OS, likely due to the success of salvage regimens and autologous haematopoietic stem cell transplantation for patients who relapse. However, with HD2000, there were higher rates of grade 3–4 neutropenia (*p* = 0.016) and associated severe infections (*p* = 0.003) in patients receiving BEACOPP when compared with ABVD. These findings of increased toxicity following BEACOPP were echoed by the LYSA H34 trial.

Avoiding relapse is a key strategy to reduce the overall toxicity burden for a patient, and a number of large trials have used iPET after 2 cycles of ABVD to stratify patients into good and poor risk, with dose intensification to eBEACOPP given to all patients who failed to achieve iPET CMR [38,39,40]. The use of consolidation radiotherapy after this strategy has varied; it was mandated in the Italian retrospective analysis [35], but was not required in either the RATHL or SWOG S0816 trials [33,34]. Although the dose intensification question was not tested in a randomised design, the longer-term relapse free survival of the non-CMR patients in SWOG S0816 was 66%, which is improved compared with historical controls and it is now UK standard practice to dose intensify ABVD-treated patients who fail to achieve CMR at iPET2 [17,41]. If this strategy is followed, data from the UK and US trials show around 75% of stage III/IV and/or IPS3and patients will achieve a durable first remission [39,42]. Noting the numbers of patients who are dose escalated based on iPET and the number of patients treated with salvage chemotherapy at relapse, when treating advanced stage patients with an initial ABVD strategy, approximately 1 in 3 patients will need treatment intensification at the interim point, at relapse or both.

Another strategy to reduce the relapse risk with ABVD has been to substitute bleomycin for the more targeted agent, brentuximab vedotin (BV) [43]. This AAVD regimen was randomised as 6 cycles against 6 cycles of ABVD for stage III/IV patients (stage II patients were excluded) in the ECHELON-1 trial [44]. Patients had an iPET2 scan, but no change in treatment was mandated based on this scan. With a median follow-up of 24.9 months, the AAVD combination showed a significantly improved PFS compared to the ABVD (82.1% vs. 77.2%; *p* = 0.04). There was no dose escalation of patients who failed to achieve iPET2 CMR in either arm, and the PFS for these poorer risk patients was clearly inferior to the CMR patients in both arms (58.2% (AAVD) and 36.6% (ABVD)) and historically inferior to the results reported for non-CMR patients in the SWOG S0816 and RATHL trials who were dose escalated. With longer follow-up of the ECHELON-1 trial, the difference in PFS between the arms had increased and there is now a statistically significant OS difference in favour of AAVD (6-year OS 93.9% vs. 89.4%) [45]. While the addition of BV improved PFS and OS, this additional drug does bring some additional toxicities such as neutropenia (58% of the AAVD group compared to 45% of the ABVD group) and 7/9 of the AAVD deaths were due to neutropenic infections. Similarly, peripheral neuropathy was reported in 67% of AAVD patients. A major toxicity advantage of AAVD is due to the removal of bleomycin from ABVD. Pulmonary toxicity and pulmonary-related deaths were significantly increased in the ABVD arm of the trial and a significant proportion of the OS difference between the arms could be accounted for by the lack of this drug-induced toxicity in AAVD-treated patients. Improving toxicity by removing bleomycin from ABVD was also explored in the RATHL trial where patients who achieved CMR after 2 cycles of ABVD were randomly allocated 4 cycles of ABVD or 4 cycles of AVD [38]. These better risk patients achieved a longer term PFS of 84.9% with no PFS inferiority between either arm. The bleomycin-free patients had fewer pulmonary adverse events and it is now a firm international standard of care to remove bleomycin from ABVD when treating advanced-stage patients who achieve iPET2 CMR after 2 cycles of ABVD. When ABVD is being used to treat early-stage HL, there is no prospective trial evidence to support the removal of bleomycin beyond cycle 2 in CMR patients, but this has become common practice in the UK [17].

Treating patients with eBEACOPP has been shown to be highly effective across all IPS subgroups, and whereas IPS grouping still predicts for inferior PFS in ABVD treated patients [42], this cannot be shown in eBEACOPP trials [24,27]. This makes the eBEACOPP approach more appealing for some clinicians when patients present with higher risk disease. However, the first remission cure rates are also extremely high for lower IPS risk patients treated with eBEACOPP and for many clinicians this regimen remains the standard of care for all advanced stage HL patients. Reducing the intensity of eBEACOPP without reducing efficacy remains a key goal for clinical researchers and reducing the total number of cycles from 8 to 6 cycles in HD15 was shown to reduce toxicity and second cancers [24]. Reducing the toxicity of eBEACOPP further has been explored in the large prospective HD18 and AHL2011 trials using different strategies [27,46]. With both trials, patients started with 2 cycles of eBEACOPP prior to iPET2. If patients achieved a CMR, they were de-escalated to 2 cycles of eBEACOPP (HD18) or 4 cycles of ABVD (AHL2011). The HD18 trial observed a 5-year PFS of 92.2% for these patients, which was non-inferior to patients who received 6–8 cycles of eBEACOPP. Approximately 75% of patients treated with 2 cycles of eBEACOPP achieved a CMR (defined by Deauville Score (DS) 1–3), meaning that 3/4 of patients could complete treatment within just 12 weeks and have a relapse risk below 10%. This shortened course of eBEACOPP was shown to reduce the number of toxicity-associated deaths, and furthermore, shortening of chemotherapy will likely provide significant psychological benefits for patients and reduce the longer-term risks associated with treatment. Within HD18, attempts to improve the outcome for patients who failed to achieve CMR were not successful. This group of patients was randomised to receive eBEACOPP with or without rituximab, but no benefit was seen for either arm. The overall 5-year PFS for iPET2 non-CMR patients was 88.3%. The AHL2011 trial similarly de-escalated treatment after 2 cycles of eBEACOPP who achieved CMR although with this trial, patients moved to 4 cycles of ABVD. Patients who failed to achieve CMR, continued with eBEACOPP chemotherapy. The CMR patients who were de-escalated had a non-inferior 5-year PFS of 85.7% and reduced toxicity relative to the standard arm who received 4 further cycles of eBEACOPP. Patients who start with eBEACOPP and achieve a CMR will therefore have the choice of completing treatment with 6 further weeks of eBEACOPP or 16 further weeks of ABVD. Patient choice at this decision point may reflect tolerance of the first 2 chemotherapy cycles and also their views on the longer-term impact of eBEACOPP on fertility and the potential additional risks from extra cycles of steroids, cyclophosphamide and procarbazine which are not components of ABVD.

Certain patients would opt for treatment with eBEACOPP but decline this treatment because of concerns for gonadal and stem cell toxicity, particularly due to the procarbazine. There is prospective randomised data from the paediatric EuroNet-PHL-C1 study showing that removing procarbazine and replacing with dacarbazine, as part of combination chemotherapy, improves multiple toxicity parameters with similar event-free survival (EFS) rates between COPP (cyclophosphamide, vincristine, procarbazine and prednisone-89.9%) and COPDAC (cyclophosphamide, vincristine, prednisone and dacarbazine-86.1%) [8]. In the UK, there is increasing use of a modified eBEACOPP regimen, termed eBEACOPDac, where the 7 days of oral procarbazine are replaced by intravenous dacarbazine 250 mg/m^2^ given on days 2 and 3. Retrospective data have now been presented from over 200 patients treated at 20 UK, Irish and French centres [47]. Accepting the limitations of retrospective data and relatively short follow-up, the protocol has been given to a high risk IPS group of patients and demonstrated comparable PFS results to the HD18 trial. When adverse event and healthcare utilisation outcomes were compared with historical outcomes for ‘real-world’ eBEACOPP patients, there is a strong suggestion that eBEACOPDac patients require fewer red cell transfusions, fewer emergency hospital admissions and women have faster and more complete restoration of menstrual periods. Ongoing mutational analysis of DNA extracted from haematopoietic stem and progenitor cells from eBEACOPP and eBEACOPDac patients will be presented at the American Society of Haematology (ASH) 2022 [48].

Finally, the use of radiotherapy for advanced HL patients has greatly reduced in recent years. Historically, adult advanced protocols would recommend radiotherapy to residual nodal disease as sized by CT [49]. Although not prospectively validated with randomised trials, international trial groups chose to remove radiotherapy if patients achieved a CMR with end of treatment PET, independent of the nodal size on CT imaging. Comparing trials showed a marked reduction in the use of radiotherapy. For instance, with HD9, 79% of patients were irradiated post eBEACOPP, but with the introduction of end-of-treatment PET imaging in HD15, only 11% were irradiated after very similar treatment regimens. No reduction in efficacy was demonstrated between the two trials and there was no excess relapse risk in patients who were spared radiation in HD15 [24]. Similarly, with the UK LY09 trial 43% of advanced-stage patients were irradiated but this dropped to 6.5% with the most recent large UK RATHL trial, with no apparent loss of efficacy between the trials [38,50]. A particularly difficult area has been whether or not patients who present with a large mediastinal mass and are managed with ABVD as part of an advanced-stage treatment strategy would benefit from consolidation radiotherapy. Subgroup analysis from the RATHL trial has suggested that these patients who achieved an iPET2 CMR and were not irradiated did not have an excess relapse risk compared with patients who presented without a bulky mediastinal mass [51]. Furthermore, an Italian trial has addressed this question by randomising these bulky mediastinal patients to receive radiotherapy or not post 6 cycles of ABVD [52]. The trial was relatively underpowered to answer this question with statistical certainty, but the data support the practice that patients who present with bulky mediastinal disease and achieve iPET CMR post 2 cycles of ABVD do not usually require consolidation with mediastinal radiotherapy. Clearly this has potential significant longer term toxicity benefits.

## 5. Conclusions and Future Directions

While there has been great success in reducing toxicity in modern-day Hodgkin Lymphoma treatment, short- and long-term toxicities persist and while chemotherapy and radiotherapy remain the foundation stones of treatment for this disease, scope for improvement will remain. We have reached the point where the efficacy of combined modality chemotherapy and radiotherapy has likely been maximised and if we are to reduce toxicity further while maintaining or improving efficacy, we will need to integrate newer drugs into our treatment algorithms. Drugs such as brentuximab vedotin and check-point inhibitors have been shown to be highly effective at treating relapsed / refractory HL and it is a clinical research priority to see whether integrating these drugs into first-line treatment can further reduce the need for radiotherapy and chemotherapy without increasing the risk of relapse. The GHSG is due to report data from HD21 in the near future, with the hope that introducing BV into a modified eBEACOPP regimen can improve both toxicity and efficacy of eBEACOPP as used in HD18. Combining pembrolizumab and nivolumab with AVD has been shown to be effective in phase 2 trials [53,54] and recruitment to potentially practice changing phase 3 trials is underway. However, while these are exciting times for HL patients and physicians, it is important to remember that new drugs that target novel pathways bring the potential for new side effects. Neuropathy is well recognised as a BV-related toxicity, but in addition, auto-immune complications are potentially challenging when patients are treated with PD-1 inhibitors [55]. Treating young patients with these drugs will need careful supervision to ensure we do not trade the known toxicities of chemotherapy and radiotherapy for potentially less well understood toxicities that might compromise a patient’s quality of life in the shorter and longer term.

## Figures and Tables

**Figure 1 cancers-14-05390-f001:**
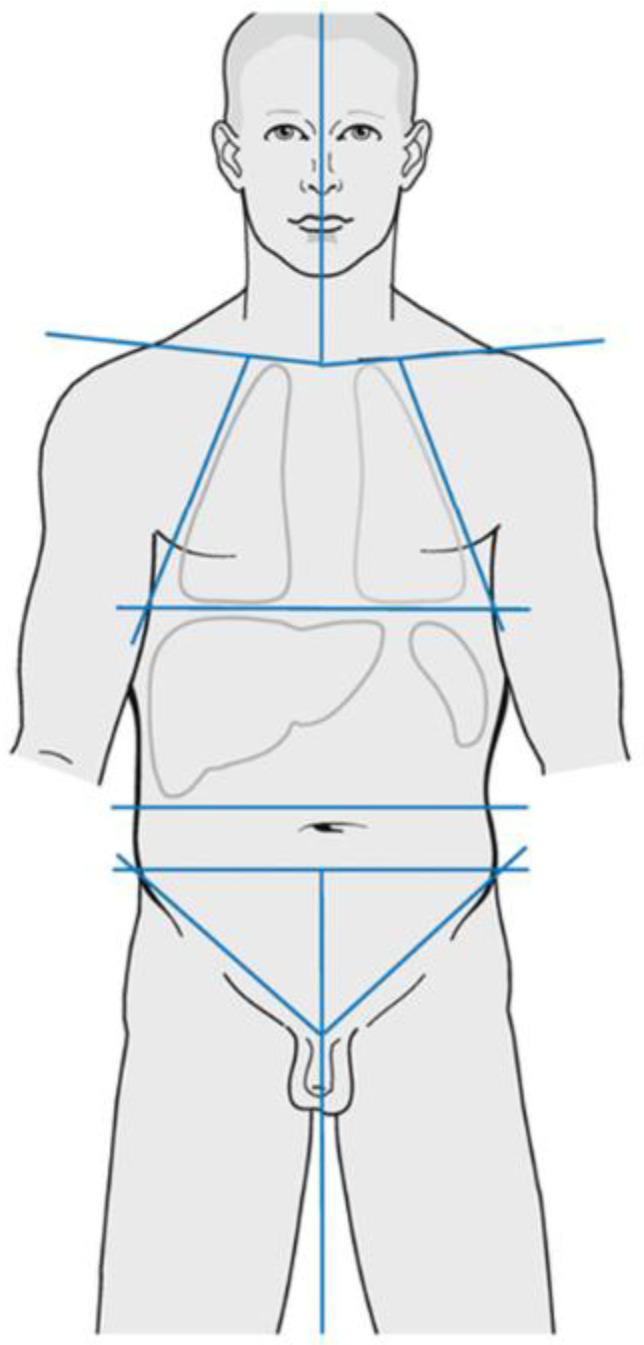
Lymph node sites as defined by the GHSG (https://en.ghsg.org/disease-stages, accessed on 3 October 2022), image taken from the BSH Guidelines for HL management [8].

**Table 1 cancers-14-05390-t001:** Risk factor definitions in Hodgkin Lymphoma. Patients classed as “unfavourable” if one or more risk factor is present. ESR, erythrocyte sedimentation rate; A, without B-symptoms; B, with B-symptoms; LP, lymphocyte predominant; NS, nodular sclerosis.

Risk Factors	GHSG	EORTC	NCIC/ECOG
Large mediastinal mass	Yes, ratio ≥ 1/3	Yes, ratio ≥ 0.35	No
Extranodal disease	Yes	No	No
Nodal areas	Yes, ≥3 areas	Yes, ≥4 areas	Yes, ≥4 areas
ESR	Yes, ≥50 (A) or ≥30 (B)	Yes, ≥50 (A) or ≥30 (B)	Yes, ≥50
Age	No	Yes, ≥50 years	Yes, ≥40 years
Histology other than LS/NS	No	No	Yes

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
