# Peer review of "Minimising the Toxicities of First Line Hodgkin Lymphoma Treatment in the Modern Era"

_cancers, 2022, doi:10.3390/cancers14215390_

Round 1

Reviewer 1 Report

This is manuscript is a very thorough and well-writen review of different treatment regimens available for adults with newly diagnosed Hodgkin Lymphoma across different risk groups. The paper discussed the different considerations about toxicities that guided choices in trial development to help reduce short and long term toxicities. Particularly helpful is when the authors try to summarize recommendations based on the trials they have discussed, recognizing that some decisions must be made on a patient by patient basis.

To help inform those decisions on a patient by patient basis I had hoped the paper would include some discussion of the specific risks of long term cardiac or pulmonary toxicity associated with anthracyclines, radiation, bleomycin, etc. For providers and patients making treatment decisions, having some understanding of the risks and benefits of treatment they are weighing beyond EFS would be helpful. It may be beyond the scope of this review but would be great to include if there was space.  

Author Response

Thank you very much for your feedback. I agree, additional information of the specific risks of each individual drug would be helpful, however we are limited by space and the need to provide an overall review of all of the different trials and guidelines for all of the different stages of HL. We have done our best to summarise the key toxicities of the main treatments and then guide readers to additional information through our references (such as the BSH guidelines). I have added some additional references now to ensure a more thorough understanding is available for those who choose to read into the details further, however in this case, extensive information on the toxicities of individual drugs is beyond the scope of this review.

Thank you again for your feedback.

Reviewer 2 Report

The present review article describes the efforts made in the recent years to reduce toxicity while maintaining efficacy in patients treated for newly diagnosed Hodgkin lymphoma. The manuscript is generally well written and there are only minor issues that should be considered in a revised version.

1) page 4: for the group of patients aged older than 60 years it has been demonstrated that ABVD should be given for a maximum of 2 cycles before switching to AVD due to the increased lung toxicity observed in later cycles. This could be mentioned briefly.

2) page 6: the results of the long-term analysis of the HD14 study could be added.

3) page 6: it should be mentioned that non-inferiority could not be demonstrated in the H10U study although treatment results between the experimental and standard arms for patients with a CMR at interim PET were very similar (André at al., J Clin Oncol, 2017)

4) page 6: with regard to novel approaches in early unfavorable stages, the results of the randomized phase II study evaluating BV-AVD in this patient group should be discussed (Fornecker et al., J Clin Oncol, 2022).

5) page 8: the full text publication reporting a 6-year OS benefit with BV-AVD in advanced stages (ECHELON-1) should be cited (Ansell et al., NEJM, 2022).

6) page 8/9: with regard to the optimization and deescalation of the BEACOPPesc protocol by implementing BV, the results of a randomized phase II study evaluating BrECADD and BrECAPP could be discussed (toxicity --> Eichenauer et al., Lancet Oncol, 2017; efficacy --> Damaschin et al., Leukemia, 2022).

7) General comment: The publication from de Vries et al., J Natl Cancer Inst, 2021 could be discussed as it illustrates that most HL long-term survivors die from cuases possibly related to their prior treatment for HL. In addition, the major publication on second malignancies (Schaapveld et al., NEJM, 2015) and  a larger publication on infertility could be discussed and added as references.    

Author Response

Thank you very much for taking the time to review the manuscript and provide comments. I have responded in the following way:

1) I have clarified that it is now common practice in the UK to treat those over 60 years with 2 x AVD (rather than ABVD) due to increased toxicities of bleomycin.

2) I have added in the long term HD14 results as food for thought and to consider when creating future guidelines.

3) I have now explicitly clarified that there was non-inferiority between groups rather than just giving the PFS data alone. 

4) I briefly mentioned the Fornecker (2022) paper and highlighted the importance of closely observing newer trials such as this, which incorporate novel agents to improve efficacy and minimise toxicity. 

5) I have ammended the citation to give the most recent results of the ECHELON-1 study. 

6) I believe this is beyond the scope of our review as we already discuss how many new and exciting trials there are, I am not sure how much a detailed review on an additional trial (especially one still in early phases) would add. But thank you very much for the recommendation- if it were a longer paper, I would definitely discuss it. 

7) I have looked at these papers and added additional citations as suggested.

Thank you again for your feedback, it has been very helpful. 

Best wishes,